# The Influence of Pulsed Electric Field and Air Temperature on the Course of Hot-Air Drying and the Bioactive Compounds of Apple Tissue

**DOI:** 10.3390/molecules28072970

**Published:** 2023-03-27

**Authors:** Agnieszka Ciurzynska, Magdalena Trusinska, Katarzyna Rybak, Artur Wiktor, Malgorzata Nowacka

**Affiliations:** Department of Food Engineering and Process Management, Institute of Food Sciences, Warsaw University of Life Sciences (WULS-SGGW), 159c Nowoursynowska St., 02-776 Warsaw, Poland

**Keywords:** pulsed electric field, plant tissue, hot air drying, chemical properties, pretreatment

## Abstract

Drying is one of the oldest methods of obtaining a product with a long shelf-life. Recently, this process has been modified and accelerated by the application of pulsed electric field (PEF); however, PEF pretreatment has an effect on different properties—physical as well as chemical. Thus, the aim of this study was to investigate the effect of pulsed electric field pretreatment and air temperature on the course of hot air drying and selected chemical properties of the apple tissue of Gloster variety apples. The dried apple tissue samples were obtained using a combination of PEF pretreatment with electric field intensity levels of 1, 3.5, and 6 kJ/kg and subsequent hot air drying at 60, 70, and 80 °C. It was found that a higher pulsed electric field intensity facilitated the removal of water from the apple tissue while reducing the drying time. The study results showed that PEF pretreatment influenced the degradation of bioactive compounds such as polyphenols, flavonoids, and ascorbic acid. The degradation of vitamin C was higher with an increase in PEF pretreatment intensity level. PEF pretreatment did not influence the total sugar and sorbitol contents of the dried apple tissue as well as the FTIR spectra. According to the optimization process and statistical profiles of approximated values, the optimal parameters to achieve high-quality dried apple tissue in a short drying time are PEF pretreatment application with an intensity of 3.5 kJ/kg and hot air drying at a temperature of 70 °C.

## 1. Introduction

Drying is one of the oldest and most commonly used food processing techniques. It consists of the exchange of heat and mass between a dried product and a drying agent, which is associated with a phase change, i.e., evaporation. Thanks to this process, the course of chemical reactions and the development of microorganisms in products can be inhibited, which extends the shelf-life of food products [1]. Although there are other advantages of drying food, there are also disadvantages, such as changes in sensory characteristics and color, as well as the degradation of nutrients susceptible to high temperatures, which are all associated with deterioration of the quality of the final product [2]. Undesirable changes that occur during the drying process can be minimized by selecting appropriate drying parameters, for example, temperature, humidity, and speed of air drying [3]. To improve the quality of the final product, new, more efficient drying methods have been developed. Hybrid drying using ultrasonics or microwaves is increasingly used [1,4]. The wide range of possibilities for combining various drying techniques have resulted in many benefits, and therefore, have contributed to the continuous development of science in this field and the improvement of existing technologies [5].

Microwave-convective drying uses the action of microwaves, i.e., electromagnetic waves, which have a frequency in the range from 300 MHz to 300 GHz. During application, microwaves are absorbed by a material, where they are converted into heat, which increases the temperature of the material. Inside the material, a higher temperature is reached in relation to the temperature outside, which accelerates heat and mass exchange [6].

Microwave-convective drying (hot air drying) is characterized by better heat transfer, which results in faster drying of a material, and the product is characterized by better sensory properties in relation to the dried material obtained using convective drying. These advantages were proven by, among others, Szadzińska and Mierzwa [7] who studied the effect of microwave-convective drying on the kinetics of drying and the quality of white mushrooms.

The goal of pretreatment is to shorten the drying time, which is associated with a reduction in energy costs during drying. In addition, applying pretreatments with properly selected parameters can effectively maintain the quality of the final material, for example, with high contents of certain nutritive compounds (anthocyanin, total phenolics, vitamin C, and antioxidant activity). Pulsed electric field (PEF) pretreatment is a type of non-thermal technology that is becoming more and more popular. Under the influence of an applied electric field of appropriate intensity, pores are formed that increase membrane permeability and facilitate the removal of water from the material during drying [8]. Due to the holes created, it is easier to transfer various components, for example, ions, to the inside of the cell [9]. PEF sets in motion ions that are located on the inside and outside of material particles. The mechanism of electroporation is related to transmembrane ion transfer. The movement of ions takes place analogously to the direction of the applied electric field. As a result, oppositely charged electric charges accumulate on each side of the cell membrane, which interact with each other, and therefore, lead to an increase in the pressure in the cell, resulting in modification of the cell membrane thickness, and then its rupture [10].

The pulsed electric field method, through the phenomenon of electroporation, significantly affects the drying process of plant origin materials [11]. Rahaman et al. [12] studied the effect of pulsed electric field as a pretreatment on the kinetics of plum drying. They found that as the intensity of the pulsed electric field increased, the drying rate of the plum and the amount of water removed increased. Mirzaei-Baktash et al. [13] studied the effect of PEF on the kinetics of convective drying of mushrooms. They showed that, in mushrooms treated with PEF, from 20 to 32% shorter drying time was needed to obtain a constant moisture content compared to a control sample. The use of PEF pretreatment resulted in obtaining dried material characterized by high contents of L-ascorbic acid and total polyphenol content, as well as high antioxidant capacity compared to the samples obtained without this treatment [14]. The selection of appropriate conditions for PEF application and the type of drying used have a significant impact on the quality of the dried material subjected to PEF pretreatment [15].

Pulsed electric field pretreatment with appropriate parameter values has a positive effect on the bioactive components of the dried material, while PEF application that is too intense (3 kJ/kg used in presented investigations) may have a negative impact on the final quality of the dried material [15]. Studies have confirmed the beneficial effect of PEF on preserving the total polyphenol content and ascorbic acid content, as well as on the antioxidant capacity of the dried material [9]. Spinach dried using hot air drying and treated with a pulsed electric field had a higher level of L-ascorbic acid and better color compared to untreated samples [14]. Mango dried by convective and vacuum with PEF pretreatment was characterized by high levels of polyphenols and flavonoids; however, the level of carotenoids was slightly reduced during drying. In addition, the application of pulsed electric field at a lower intensity was more beneficial, which may have been due to oxidation of carotenoids that were sensitive to the presence of air or free radicals formed during pulsed electric field treatment [15].

The aim of the work was to investigate the impact of pulsed electric field and air temperature on the course of hot air drying, and the selected properties of apple tissue. Apple tissue was subjected to different intensity levels for the parameter pulsed electric field intensity (1, 3.5, and 6 kJ/kg), and then hot air drying at three different temperatures (60, 70, and 80 °C). The following properties were determined: water activity, total polyphenol content, flavonoid content, antioxidant activity based on the degree of quenching of the synthetic DPPH radical and reducing power in the obtained dried apple tissue, vitamin C content, sugar content, and characteristic bonds between molecules according to Fourier transform infrared spectroscopy (FTIR).

## 2. Results

### 2.1. The Influence of PEF on the Kinetics of Drying Apple Tissue

Figure 1 shows the drying time of apple tissue subjected to different PEF pretreatment intensity levels and dried at different drying air temperatures in relation to the relative water content (humidity ratio, MR). The shortest drying time was characterized by the sample marked as PEF 6, 70 °C, with an average drying time of 120 min, which was about 37% shorter than the control sample marked as Control 70 °C. The longest drying time was obtained for the sample marked as Control 60 °C. In this case, the time needed for the complete evaporation of water was equal to 220 min. The air temperature of 80 °C resulted in faster drying of the apple tissue compared to other temperatures. The response surface results of the drying time showed a good fit for the model. According to Figure 2, it can be seen that the dried material is obtained in a shorter drying time when a higher temperature is used, while the PEF pretreatment does not affect it significantly. Whereas a longer drying time is obtained when the lower temperature is applied.

### 2.2. Influence of PEF on Water Activity of Dried Apple Tissue

The fresh apples were characterized by high water activity of 0.986 (Table 1). The drying resulted in a high reduction in water activity.

The dried apple tissue was characterized by water activity values ranging from 0.211 to 0.331. Figure 3 shows the water activity of the dried apple tissue obtained using the different values of the selected parameters. The water activity values were affected by temperature and the interaction of PEF pretreatment with temperature, with a greater effect of temperature (η^2^ = 0.93), as shown by the two-factor analysis of variance. The lowest water activity, equal to 0.211, was found in the dried apple tissue obtained after PEF pretreatment with an electric field intensity level of 6 kJ/kg and dried using the hot air drying temperature of 70 °C. The highest water activity was found in the the dried apple tissue subjected to PEF pretreatment with an electric field intensity level of 6 kJ/kg but dried at the temperature of 60 °C. Based on the homogeneous groups, it can be concluded that, in the most cases, there were significant differences between the water activities in the obtained dried samples. Since the response surface results of the water activity were not a good fit for the model, this figure was not shown.

### 2.3. Influence of PEF on Total Polyphenol Content and Flavonoid Content in Dried Apple Tissue

Based on the results of the conducted research, it can be observed that the total polyphenol content in the dried apple tissue was affected by both the electric field intensity level (η^2^ = 0.83) and the drying temperature (η^2^ = 0.78) as well as the interaction between these two parameters (η^2^ = 0.93). Figure 4 shows changes in the total polyphenol content of dried apple tissue depending on the applied electric field intensity level and the temperature. The total polyphenol content in the fresh material was 1682.9 mg of chlorogenic acid per 100 g of dry substance (ChlA/100 g d.m.). At a temperature of 60 °C, the total polyphenol content was equal to 1277.38 mg ChlA/100 g d.m. PEF pretreatment at each intensity level resulted in a decrease in the total polyphenol content compared to that of the dried material obtained without the use of pretreatment. With an increase in the intensity level of PEF, degradation of polyphenols in the apple tissue was observed. For PEF pretreatment with an electric field intensity of 1 kJ/kg, the total polyphenol content was the highest (849 mg ChlA/100 g d.m) compared to those of other dried fruits obtained with prior pretreatment, which was about 33% lower than the value of apples not treated with PEF. Increasing the PEF pretreatment electric field intensity levels to 3.5 and 6 kJ/kg made it possible to obtain dried fruit with total polyphenol contents of 844.9 and 790.4 mg ChlA/100 g d.m., respectively. This means that the result was about 34% and 38% lower as compared to the dried fruit obtained with a PEF pretreatment electric field intensity of 1 kJ/kg.

When drying at 70 °C, it can be concluded that, as in the case of drying at 60 °C, the use of the pretreatment reduced the number of polyphenols compared to the control samples (dried apple tissue obtained without pretreatment). In the case of the pretreatment, the highest total polyphenol content was equal to 1201.1 mg ChlA/100 g d.m., which was found in the dried material obtained as a result of applying a pulsed electric field intensity of 1 kJ/kg. This result is about 19% lower than the material that was not treated with PEF. Increasing the pulsed electric field intensity resulted in a further decrease in total polyphenol content. For PEF pretreatment with an electric field intensity of 3.5 kJ/kg, the total polyphenol content is about 34% lower than the result at the lowest PEF pretreatment electric field intensity, and at 6 kJ/kg, this value is 40% lower than the result at a PEF pretreatment electric field intensity of 1 kJ/kg.

Analyzing the results obtained when the drying temperature was at 80 °C, it can be seen that an increase in the PEF pretreatment intensity resulted in a slight increase in the total polyphenol content in the apple tissue; however, it should be highlighted that the highest value for total polyphenol content was found in the dried apple tissue obtained without pulsed electric field pretreatment. For example, apple tissue pretreated with a PEF pretreatment electric field intensity of 1 kJ/kg was characterized by a total polyphenol content that was 42% lower than the reference sample. Increasing the PEF pretreatment electric field intensity to 3.5 kJ/kg resulted in an increase in the retention of phenolic compounds by approximately 22.3 mg ChlA/100 g d.s. compared to that of the dried apple tissue obtained at an electric field intensity of 1 kJ/kg. In the case of the drying temperature of 80 °C, a pulsed electric field pretreatment with an electric field intensity of 6 kJ/kg was the most advantageous procedure, since the dried apple tissue reached the highest total polyphenol content among the pretreated variants, i.e., equal to 882.7 mg ChlA/100 g d.m. However, this value was about 35% lower than the result obtained in the dried apple tissue obtained without pretreatment.

Figure 5 shows the response surface results of the total polyphenol content of dried apple tissue after applying pulsed electric field at different intensity levels and then hot air drying at different temperatures. The statistical analysis showed a good fit for the model. According to the model, the highest value of the total polyphenol content was obtained by applying the PEF pretreatment with the lowest electric field intensity (1 kJ/kg) and a drying temperature of 70 °C.

Similar dependencies of the effect of PEF pretreatment on the total flavonoid content in dried apple tissue were shown as in the case of the analysis of total polyphenol content (Figure 6). The two-factor analysis showed that both parameters, i.e., PEF pretreatment intensity (η^2^ = 0.94) and the temperature of drying (η^2^ = 0.73), as well as the interaction between the parameters (η^2^ = 0.96) had a significant impact on the total flavonoid content. It was found the PEF pretreatment caused a decrease in the total flavonoid content, and that increasing the PEF energy input, in most cases, caused a slight decrease in the analyzed results. While the drying temperature had no significant effect on the total flavonoid content.

Figure 7 shows the response surface results of the total flavonoid content of dried apple tissue obtained as a result of pulsed electric field pretreatment at different intensity levels and hot air drying in a convective dryer at different temperatures. The statistical analysis showed a good fit for the model. In addition, it showed that the highest value of the total flavonoid content was obtained for apple tissue subjected to a PEF pretreatment with an electric field intensity of 1 kJ/kg and dried at 70 °C.

### 2.4. The Influence of PEF on the Vitamin C Content in Dried Apple Tissue

Figure 8 shows the change in vitamin C content in dried apple tissue. The statistical analysis showed that the change in vitamin C content in the dried apple tissue was affected by both the pulsed electric field intensity level (η^2^ = 0.97) and the drying temperature (η^2^ = 0.90) as well as the interaction between both of these parameters (η^2^ = 0.91). The two-factor analysis showed that the PEF pretreatment had a more significant effect on the content of vitamin C. The pretreatments caused a decrease in the amount of ascorbic acid compared to the fresh apple sample with a vitamin C content equal to 143.2 mg/100 g d.m. The content of ascorbic acid ranged from 130.6 to 6.5 mg/100 g d.m. In each of the variants, there was a tendency of decreasing ascorbic acid content caused by the application of pulsed electric field pretreatment. Taking into account all drying temperatures and PEF pretreatment intensities, the highest vitamin C content (19.7 mg/100 g d.m.) in dried apple tissue treated with an electric field was found in Sample PEF 1, 70 °C and the lowest content (6.5 mg/100 g d.m.) in dried apple tissue was found in Sample PEF 6, 60 °C. The conducted one-factor analysis of variance showed that the tested samples differed significantly based on the examined parameter.

Figure 9 shows the response surface results of vitamin C content in dried apple tissue obtained as a result of applying pulsed electric field at different intensity levels and hot air drying at different temperatures. The statistical analysis showed a lack of fit of the model in statistical terms, thus, this figure was not shown.

### 2.5. The Influence of PEF on the Antioxidant Activity of Dried Apple Tissue

Table 2 shows the change in the antioxidant capacity of dried apple tissue obtained by applying PEF pretreatment at different intensities and hot air drying at different drying temperatures. The antioxidant capacity was determined taking into account the degree of quenching of the synthetic DPPH radical, presenting the results as the content of mg of trolox per 1 g of dry substance (mg TE/g d.m.) as well as the iron ion reducing power expressed in mg of trolox per 1 g of dry substance (mg TE/g d.m).

The parameters used for processing reduced the antioxidant capacity of the dried apple tissue compared to fresh apple tissue, whose antioxidant capacity was 4.33 mg TE/g d.m. (see Table 2). The antioxidant activity evaluated with DPPH radicals for the obtained dried apple tissue ranged from 2.26 to 3.8 mg TE/g d.m. The highest antioxidant capacity was shown by the dried apple tissue Sample Control 70 °C, and the lowest antioxidant capacity was shown by Sample PEF 6, 70 °C. Taking into account all the drying temperatures, the application of each of the selected electric field intensities resulted in a decrease in the content of mg TE/g d.m. compared to the control samples (dried samples obtained without pretreatment). The smallest loss of antioxidant activity compared to the control samples was obtained in the case of apple tissue dried at 70 °C and PEF pretreatment with an electric field intensity of 1 kJ/kg, where the antioxidant capacity only decreased by 6.3% and this change was not statistically significant. The conducted two-factor analysis of variance showed that the PEF pretreatment intensity (η^2^ = 0.74) and the interaction between PEF pretreatment and temperature (η^2^ = 0.84) had a significant impact on the antioxidant capacity of the tested material.

Similar results were obtained for reducing power (RP) determination (Table 1). Samples PEF pretreatment applied obtained significantly lower reducing power than apple tissue dried without pretreatment. An increase in the PEF pretreatment intensity, in most cases, had a decreasing effect but generally it was statistical insignificant. However, in the case of both parameters: PEF pretreatment intensity (η^2^ = 0.97), temperature (η^2^ = 0.75), and the interaction between the parameters (96) had significant effects on the RP. Furthermore, Figure 9a,b show the response surface results of the antioxidant capacity with DPPH radicals and reducing power for dried apple tissue obtained with different parameter process values. The statistical analysis showed a good fit for the model for both parameters.

### 2.6. The Influence of PEF on the Content of Sugars in Dried Apple Tissue

Figure 10 shows changes in the total sugar content including sucrose, glucose, fructose, and sorbitol in apple samples that were hot air dried at different temperatures after PEF pretreatment at various intensity levels. The total sugar content in dried apple tissue varied from 24.29 g /100 g d.m. (for PEF 3.5, 60 °C) to 37.34 g d.m. (for PEF 1, 60 °C), whereas the total sugar content in the fresh apples was equal to 36.71 g/100 g d.m. For almost all tested samples (except for apple tissue dried at 60 °C and PEF pretreatment with an electric field intensity of 1 kJ/kg) the total sugar content decreased as a result of the applied treatment. When the electric field intensity was 1 kJ/kg, the total sugar content was higher than in the case of the PEF pretreatments with electric field intensities of 3.5 or 6 kJ/kg, regardless of the drying temperature. For all samples, on the one hand, fructose accounted for the highest share of the total sugar content, ranging from 62.87 to 69.71%. On the other hand, sorbitol accounted for the lowest share of the total sugar content, ranging from 1.49 to 2.67%. Furthermore, glucose and sucrose accounted for between 14.48 and 21.99% and between 6.93 and 19.98% of the total sugar content, respectively. Additionally, increasing fructose and glucose content was noted with an increase in the drying temperature for samples not subjected to PEF pretreatment. The fructose content for apple tissue dried at 60, 70, and 80 °C was 16.54, 19.85, and 22.23 g/100 g d.m., respectively, and the glucose content was 4.29, 4.73, and 6.07 g/100 g d.m., respectively. On the contrary, with the higher drying temperature, sucrose content was lower or constant and it was equal to 4.01, 3.54, and 3.54 for apple tissue dried at 60, 70, and 80 °C, respectively. Sorbitol content was the lowest among the no PEF pretreated samples when convective drying was carried out at a temperature of 70 °C (0.44 g/100 g d.m.); both an increase and a decrease in drying temperature caused an increase in sorbitol content in the analyzed samples. The statistical analysis showed that both sucrose content and sorbitol content did not differ significantly among the tested samples but some significant differences in glucose content and fructose content were observed among the samples. 

The two-factor analysis showed that both parameters (PEF pretreatment intensity and drying temperature) and the interaction between them significantly influenced the total sugar content as well as sucrose, fructose, glucose, and sorbitol contents.

### 2.7. The Influence of PEF on the Chemical Properties of Dried Apple Tissue (FTIR Method)

Figure 11 shows the FTIR spectra of a dried apple tissue. The spectra reflect the correlations between the absorbance intensity of the radiation and its energy described as the wave number (cm^−1^). Six peaks of the IR spectrum are listed. Each of the obtained dried samples showed a similar pattern of spectra.

## 3. Discussion

PEF pretreatment is usually used to improve the drying process. In our study, we found that, in most cases, after applying PEF pretreatment, the time needed for complete evaporation of water from apple tissue was shortened along with an increase in the pulsed electric field intensity. Such results can be explained by electroporation during PEF pretreatment. Electroporation is the formation of pores in tissue, which facilitates the process of water evaporation from a material [8]. Wiktor et al. [16] argued that the cell disintegration index increased with both an increase in electric field intensity and in pulse numbers for apple tissue air dried with PEF pretreatment. As the intensity of the electric field increases, the number of pores increases, and therefore, the average drying time of the product is shortened. Ostremeier et al. [17] dried onion tissue at three drying air temperatures, i.e., 65, 75, and 85 °C, and studied the impact of PEF pretreatment (PEF pretreatment with an electric field intensity of 4 kJ/kg) on the time of convective drying of plant tissue. Their research showed that the use of PEF pretreatment shortened the drying time by an average of approx. 25% compared to the control samples (without PEF application). Chauhan et al. [18] studied the effects of PEF pretreatment intensity, duration of the pulsed electric field, and the water temperature during pretreatment on the drying time of apple slices. The experiment showed that an increase in the electric field intensity accelerated the apple drying process due to the increasing degree of disintegration of plant tissue cells. A statistical analysis of the obtained results proved that, in the case of the combined application of the mentioned process parameters, the number of pulses used and the water temperature during the pretreatment had more significant impacts on the time of the apple drying process. This was most likely due to the fact that the applied electric field intensity was relatively low and did not cause sufficient disintegration of apple tissue cells, which led to the acceleration of mass exchange. The increasing number of delivered impulses meant that the investigated material was subjected to pretreatment for a longer period of time, and thus, stayed in hot water longer, which also strongly influenced the breakdown of apple tissue cells, and therefore, resulted in a noticeable shortening of the duration of the apple drying process.

Removing water from food extends its shelf-life, which, at the same time, extends the possible storage time. It is necessary to select the appropriate parameters of the drying process in order to obtain dried material of the best quality in terms of, for example, the content of nutrients and chemical ingredients [19]. Water availability in plant material is characterized by the concept of water activity, which affects the chemical processes taking place, as well as the limited development of microflora in food products. It has been reported that water activity below 0.6 prevented the development of microorganisms [20]; each of the dried materials had a water activity below 0.6, which allowed for inhibition of the growth of microorganisms in the material. It has been shown that temperature has an important effect on water activity, and the use of PEF pretreatment improves this effect. Ostremeier et al. [17] showed that samples, in which PEF pretreatment was applied, were obtained that had lower residual moisture even if a lower temperature of drying was used. The authors supposed that PEF pretreatment not only facilitated mass transfer in the first drying stage but also improved the second drying stage, which was characterized by the migration of moisture from the inner interstices of the sample to the outer surface.

Polyphenols are compounds found in food, which belong to the group of bioactive ingredients that are important for the human body. The content of these compounds affects the quality and antioxidant properties of food products [21]. The degradation of bioactive compounds can be influenced by numerous technological processes, for example, drying, extrusion, pasteurization, sterilization, and the use of preliminary treatments, including pulsed electric field pretreatment [22]. In the conducted research, we showed that the content of polyphenols and flavonoids in the dried apple tissue was affected by both the supplied electric field intensity and the drying temperature. The use of PEF pretreatment caused a decrease in total polyphenol content and total flavonoid content compared to the control samples (dried without PEF application). An increase in PEF pretreatment intensity was accompanied by a decrease in the total polyphenol content in the dried apple tissue. Changes in total polyphenol content depend on the efficiency of electroporation, measured, for example, by determining the degree of disintegration. A PEF pretreatment intensity that is too high can lead to the degradation of bioactive compounds, and properly selected values for the parameters can improve the extractability of these compounds. Similar relationships were observed by Wang et al. [23], who studied the impact of pulsed electric field treatment on the extraction of bioactive ingredients from apple skins. Mello et al. [9] indicated that pulsed electric field pretreatment before the drying process positively affected preservation of the total polyphenol content, ascorbic acid, and the antioxidant capacity of the dried material. Additional benefits can be achieved by the combination of pulsed electric field pretreatment with hybrid drying. Orange peel dried with PEF pretreatment has been characterized with a higher total polyphenol content in relation to orange peel dried in a hybrid way without pretreatment. In addition, Lammerkitten et al. [15] showed that PEF pretreatment had a positive effect on the contents of phenols and flavonoids in dried mango, which was associated with the phenomenon of electroporation that increased the permeability of the cell membrane and consequently improved the extractability of phenolic compounds. They showed that the use of PEF pretreatment with a lower specific intensity was the most beneficial. Lammerskitten et al. [24] showed that the total phenolic content was increased by up to 47% for PEF pretreated freeze-dried apple tissue compared to untreated freeze-dried apple tissue.

Vitamins are a large group of organic chemical compounds that are characterized by diverse structures. Their task is to support the proper functioning of the body. They belong to the group of exogenous compounds, for example, compounds that the human body is unable to produce, and therefore, it is necessary to supply them through food intake. Vitamin C is a bioactive compound and has strong antioxidant properties, which help to prevent heart and capillary diseases. Ascorbic acid can also be used as a food additive. Due to its good antioxidant properties, it slows down oxidation reactions, which allows products to extend their shelf-life [25,26]. Yamakage et al. [14] studied the effect of pulsed electric field pretreatment on changes in the quality of spinach during hot air drying and found that the use of PEF pretreatment obtained a dried product with a high content of L-ascorbic acid compared to the dried product obtained without the PEF pretreatment. The intensity of the applied PEF pretreatment had a greater influence on the content of vitamin C. A decrease in the amount of ascorbic acid was found in relation to the control sample. Ascorbic acid is very susceptible to degradation under the influence of the drying process. The use of PEF pretreatment reduced the acid content of the orange peel compared to the control sample. This relationship was most likely due to the fact that the greatest degradation of ascorbic acid was caused by drying itself. Pulsed electric field pretreatment leads to the electroporation process, which increases the exposure of nutrients to hot air during drying, and therefore, contributes to the degradation of ascorbic acid [9]. The effect of pulsed electric field pretreatment was used by Morales-de la Peña et al. [27], who studied the effect of high-voltage PEF pretreatment on changes in the content of vitamin C in stored fruit juices. The results were compared to the content of ascorbic acid in juices not subjected to PEF pretreatment and to the fruit juices subjected to thermal treatment (90 °C). On the basis of vitamin C content, studies have shown that a shorter duration of PEF pretreatment is more beneficial. However, regardless of the duration time of PEF pretreatment, samples subjected to PEF pretreatment retained a higher amount of vitamin C compared to juices treated at a high temperature, but each treatment resulted in a greater decrease in the amount of vitamin C versus the control (untreated). Since PEF pretreatment is a non-thermal treatment, the higher content of vitamin C in juices with electric field obstruction may result from the low resistance of ascorbic acid to the high temperature used during thermal treatment. With PEF pretreatment, greater degradation of vitamin C in the juice obtained after a longer duration pretreatment may be due to the increase in the temperature of the material; the juice was more intensively subjected to high-voltage electricity, which caused the Joule effect, resulting in an increase in the temperature of the material, which resulted in a higher degree of degradation of ascorbic acid. When comparing the results of these studies and those presented in this paper, one should bear in mind the differences between the compared processes and the form (“state of aggregation”) of the tested matrices.

Antioxidants are chemical compounds that belong to the group of bioactive ingredients present mainly in fruits and vegetables. One of the characteristics of biologically active compounds is that living organisms are not able to produce them, and therefore, they should be supplied through food intake. The role of antioxidant substances is to protect cells and tissues against the adverse effects of emerging free radicals (unpaired oxygen atoms), which cause undesirable changes in the human body. Antioxidants inhibit the action of free radicals by “connecting” with unpaired electrons, slowing down the oxidation reactions taking place in the body. The antioxidant compounds include polyphenols, flavonoids, vitamin C, carotenoids, and xanthophylls [28,29]. The conducted analysis showed that the applied PEF pretretment intensity, in most cases, had an insignificant impact on the antioxidant capacity of dried apple tissue and that PEF pretreatment caused loss of antioxidant activity compared to the control samples (dried without PEF application). Lammerskitten et al. [24] also showed that, for freeze-dried apple tissue, PEF pretreatment decreased antioxidant activity up to ∼60% compared to a reference sample. A different effect of PEF pretreatment on the antioxidant properties of dried material was shown by Huang et al. [30], who applied PEF PREtreatment to fresh apricots. The use of a higher electric field intensity improved the antioxidant capacity of the dried material. The differences in the results could have been influenced by the sodium sulfite used, which has an oxidizing capacity. The combination of these two factors enhanced the ability to reduce free radicals in dried apricots. Mello et al. [9] showed that the dried material obtained with PEF pretreatment had a higher antioxidant capacity for a shorter application time and this activity was similar to the dried material obtained using hybrid drying without pretreatment. Reducing the antioxidant capacity when using higher intensity values of the PEF pretreatment parameter may be caused by greater exposure of bioactive cells to high temperature during drying, which leads to a decrease in the value of bioactive compounds and the antioxidant capacity of the dried material. Therefore, PEF pretreatment with appropriate parameter values positively affects the quality of the dried material, while too intensive PEF pretreatment may have a negative impact on the final quality of the dried material.

Reducing power has also been used to evaluate the ability of natural antioxidants in dried apple tissue pretreated using PEF and the obtained results correlated with the polyphenol and flavonoid contents. In addition, Yakubu et al. [31] showed a significant decrease in the reducing power and the DPPH scavenging activity of blanched bitter leaves. The increase or decrease in antioxidant activity was explained by Adefegha and Oboh [32] by the fact that heat treatments can soften the matrix and can improve or degrade the extractability of phytochemicals involved with antioxidant activity.

Sugars are essential in the human diet for proper body functioning. Fructose and glucose are important monosaccharides. Chemically combined, these sugars form a disaccharide, i.e., sucrose. Such sugars as glucose, fructose, and sucrose are present, among others, in fruit and vegetables. Fructose supplies the human body with quick energy, whereas glucose is necessary as one of the primary sources of fuel for cellular metabolism. Furthermore, sorbitol belongs to polyols, which are saccharide derivatives. Compared with sugars, polyols, including sorbitol, are poorly absorbed, and therefore provide fewer calories and lower glycemic responses [33,34]. Regarding sugars, apple tissue contains mostly fructose, but certain amounts of glucose and sucrose are usually also present in these fruits, whereas sorbitol is present in apple tissue in a much smaller quantity [35,36,37]. This was also confirmed by our study (Table 2). The total sugar content in apple tissue as well as specific sugar contents (i.e., fructose, glucose, and sucrose) vary depending on the variety, weather conditions, culture technology, and also the position and exposition of the apple tissue in the crown. The specific sugar content during storage may both decrease or increase, which is also determined by the apple variety [38]. Wojdyło et al. [39] studied the influence of different drying methods and conditions on the quality parameters of red-fleshed apple fruit snacks, including the total sugar content. The authors observed that sugar content was strongly determined by the drying method. The samples obtained by freeze-drying were characterized by the highest sugar content which was 34.10 g/100 g d.m., whereas the sugar contents of convective dried samples were 24.3 ± 2.5, 26.7 ± 3.1, and 21.7 ± 1.8 g/100 g d.m. depending on the drying temperatures which were 50, 60, and 70 °C, respectively. These results were similar to the results reported in this research. When analyzing the apple tissue samples without PEF pretreatment, the increasing contents of both fructose and glucose together with the decreasing or constant content of sucrose was observed with an increase in the drying temperature. Similarly, Delgado et al. [40] reported the phenomenon of increased fructose content during convective drying of chestnuts, while Mitrović et al. [41] reported a decrease in sucrose content and, simultaneously, an increase in inverted sugars content after convective drying of plums at 70 or 90 °C. Furthermore, Macedo et al. [42] studied the effect of convective drying temperature (40, 60, and 80 °C) on the drying kinetics and the physicochemical properties of dried bananas, and also observed that the reduced sugar content of the dried fruits was higher as the drying air temperature increased. The authors explained that the results were probably related to sucrose hydrolysis to glucose and fructose during the convective drying [40,41,42]. In the present research, the effect of PEF pretreatment was not clear and it varied depending on the PEF pretreatment intensity applied and also the temperature of the following hot air drying. The content of sucrose did not differ significantly but statistically significant differences in both glucose and fructose content between the samples were noted, which was in accordance with the research of Rybak et al. [43], in which the influences of various treatments, including PEF pretreatment with an electric field intensity of 1 or 3 kJ/kg, on the selected properties of red bell pepper were analyzed. Additionally, the authors suggested that the increase in the glucose content of PEF-treated samples might be caused by the decomposition of carbohydrates in such a way that it can affect other carbohydrates, and also improve the extractability of sugars [43]. Furthermore, PEF pretreatment of chokeberry juice sources from six different farms resulted in an increase in the total sugar content in all cases [44]. In turn, Rybak et al. [45] recorded that PEF pretreatment with an electric field intensity of 1 or 3 kJ/kg conducted before convective or microwave-convective drying affected the significant decrease in total sugar content in red bell pepper samples. It was explained by the fact that the electroporation occurring as a result of the PEF pretreatment that caused a rupture in the cell membrane which resulted in sugar leakage.

Infrared spectroscopy (FTIR) is a method used to determine the structure of particles as well as the composition of molecular mixtures. Infrared radiation is absorbed in frequencies related to the vibration energy of the bonds between atoms in a molecule [46]. In this study, it was shown that the obtained dried apple tissue showed a similar pattern of spectra, which may have resulted from the same structure and chemical composition of the tested samples, and at the same time from the presence of the same functional groups in the tested material. According to the literature, the range of peaks from 1200 to 1500 cm^−1^ refers to the vibrations of COH, CCH, and COH bonds; additionally, the range from 1630 to 1680 cm^−1^ corresponds to the amide functional groups associated with the CO carbonyl group. The peaks at the wave values between 1149 cm^−1^ and 1336 cm^−1^ indicate the presence of COC glycosidic bonds for pectin. The IR spectra in the ranges of 3200–3500 cm^−1^ and 2900–2920 cm^−1^ are related to the stretching vibrations of the OH, NH, and CH bonds. The range associated with the CH group mainly relates to the presence of these bonds in the cellulose and hemicellulose present. The absorbed light at the value of 1710 cm^−1^ is related to the absorption of the carbonyl group of fatty acids that are present in the fibers of the raw material. In addition, the CO bonds present in lignin are absorbed at the peaks of 1634 cm^−1^ and 1374 cm^−1^. Bioactive ingredients such as phenolic compounds are identified at 1560 cm^−1^ and 1630 cm^−1^. With regard to the presence of sugars, the peak near 922 cm^−1^ is related to the α-anomeric linkage between glucose and fructose in sucrose; additionally, the glycosidic linkage in sucrose COC is about 990 cm^−1^. In fructose, the vibrations corresponding to the stretching of CO and CC bonds are at the wave of 1046 cm^−1^, in the case of sucrose, the stretching of CO bonds takes place at 1048 cm^−1^, and the value of 1102 cm^−1^ is the stretching of CO and CC bonds in glucose, as well as bending of COH bonds. The wavelength range from 896 cm^−1^ to 900 cm^−1^ is associated with β-glucosidic bonds in hemicellulose and cellulose [47,48,49]. The influence of PEF p retreatment on the chemical composition of plant tissue was studied by Ahmed et al. [50]. The research was carried out using the juice of wheat plants. PEF pretreatment with an intensity of 9 kV/cm was applied. In addition, sonication and the combination of PEF + US were used to determine the difference in the effect of pretreatment on the material properties. Similar IR spectra with similar wave values were obtained for all samples. In addition, Niu et al. [51] obtained similar results for freeze-dried naringin in pomelo tissue. Several PEF pretreatment intensities were used. Most of the peaks obtained, as well as those obtained for the control samples not subjected to PEF, coincided, and therefore, it was concluding that PEF pretreatment had no significant effect on the absorption of infrared light.

## 4. Materials and Methods

### 4.1. Materials

For this study, the the Gloster apple variety was selected from an organic farm (Warsaw, Poland). The apples were cut into 5 mm thick slices, and then they were cut into four parts, using a sharp knife. Before the apples were cut and dried, a pulsed electric field (PEF) pretreatment was applied. The properties of the fresh apple tissue are presented in Table 2.

### 4.2. Technological Methods

#### 4.2.1. Pulsed Electric Field Application

Before the drying process, the tested material was pretreated by applying a pulsed electric field (PEF) in an ELEA Pi-lot-Dual pulsed reactor (Elea Vertriebs- und Vermarktungsgesellschaft mbHVer, Quakenbrück, Germany). The PEF device has stainless steel electrodes, where the gap between them is 28 cm. The device delivers a 2 Hz frequency of exponential decay pulses with a monopolar signal and a width of 40 ms.

The treatment consisted of placing whole apples (approx. 150 g) in the chamber, and then adding tap water at room temperature in such an amount that the mass of the entire system was approx. 1000 g. The electric field intensity during the PEF application was 1 kV/cm, and the energy values delivered to the system during pretreatment were 1, 3.5, and 6 kJ/kg.

#### 4.2.2. Hot Air (Convective) Drying

The material was dried using the convective method in a prototype laboratory dryer (Promis-Tech, Wrocław, Poland) at the drying air temperature of 60, 70, and 80 °C. The air flowed through the material perpendicularly to the screen and its velocity was 1.2 m/s. Before putting the raw material into the dryer, the material was placed on a sieve in the amount of about 150 g, and then the parameters were set and the drying time was recorded every 5 min using a scale that automatically registered weight changes.

### 4.3. Analytical Methods

#### 4.3.1. Determination of the Kinetics of the Drying Process

The kinetics of drying are presented in the form of drying curves, which show the relationship between the relative water content (MR) and the progress of the process [52]. The relative water content (humidity ratio, MR) was calculated based on the following relationship:(1)$$\mathrm{MR}=\frac{u_{\tau }}{u_{0}},$$
where u_0_ is the water content in raw material (kg H_2_O/kg d.m.) and uτ if the water content of the material at the time of drying (kg H_2_O/kg d.m.).

#### 4.3.2. Determination of the Water Content

About 0.2 g (with an accuracy of 0.0001 g) of dried material, homogenized using an analytical mill, was weighed in previously dried weighing bottles, with an accuracy of 0.0001 g. Then, the vials were dried in a laboratory dryer at 70 °C for 24 h [53]. After this time, the samples were placed in a desiccator and kept until they reached room temperature. Then, the cups were weighed with an accuracy of 0.0001 g and the content of dry matter in apple tisssue was calculated. The assay was performed in duplicate.

#### 4.3.3. Determination of Water Activity

The water activity measurement was carried out in two repetitions for each type of dried material. The Aqualab 4 TE (Decagon, Pullman, WA, USA) water activity meter was used for the measurement. The measurement was made at a temperature of 25 °C [54].

#### 4.3.4. Determination of Total Polyphenol Content (TPC)

The determination was carried out using the spectroscopic method based on a color reaction of the analyte with the Folin Ciocalteau (F-C) reagent [55]. The material was ground using an analytical mill (IKA A11 basic, IKA-Werke GmbH & Co., Staufen, Germany).Then, 0.3 g of dried material was weighed into the test tube with an accuracy of ±0.0001 g and diluted with 10 mL of 80% ethyl alcohol. The solution was shaken on an orbital shaker (Multi Reax, Heidolph Instruments, Schwabach, Germany) at 20 °C and after 12 h it was centrifuged (MegaStar 600, VWR, Leuven, Belgium) at 4350 rpm for 2 min. Such extract was used for the further analysis.

In a well of a 96-well plate, 10 µL of the supernatant was placed, which was diluted twice with distilled water. Then, 40 µL of F-C reagent (5 times diluted with distilled water) was added to the solution, and after 3 min, 250 µL of 7% sodium carbonate solution was added. The samples were incubated for 60 min at 20 °C and protected from light. Absorbance was measured at a wavelength of 750 nm on a plate reader (against reagent blank). The result was expressed in mg/100 g of dry matter of chlorogenic acid using a calibration curve for chlorogenic acid standard at a concentration of 0–100 µL. The assays were performed in duplicate for each extract.

#### 4.3.5. Determination of Total Flavonoid Content

The method with aluminium (III) chloride was used to determine the total flavonoid content [56]. First, 20 µL of the extract was diluted with 80 µL of distilled water and mixed with 10 µL of NaNO_2_ (5% *w*/*v*). After 5 min, 10 µL of AlCl_3_ (10% *w*/*v*) was added and mixed, and after 6 min, 40 µL of 1 M NaOH solution was added and mixed. After 20 min, the absorbance of the solutions was measured at 510 nm. The quantitative content of flavonoids was calculated based on a calibration curve for quercetin in the range of 0–500 µg/mL. Measurements were made in duplicate.

#### 4.3.6. Determination of Antioxidant Capacity (AC) on the Basis of Free Radical Scavenging DPPH and Ferric Antioxidant Reducing Power (RP)

The antioxidant capacity was determined using a spectrophotometric method to determine the degree of quenching of the synthetic 2,2-diphenyl-1-picrylhydrazyl (DPPH) radical [57]. In order to prepare the initial DPPH solution, 0.025 g of 2,2-diphenyl-1-picrylhydrazyl was diluted to 100 mL with 99% methanol. The solution was stored for a minimum of 24 h protected from light at 4 °C to generate the radical. The working solution was prepared immediately before analysis. First, 9 mL of the starting solution was diluted with 80% ethanol solution, in which absorbance measured at 515 nm was in the range of 0.700 ± 0.020. Measurements were made in 96-well plates. Then, 250 μL of the radical solution was added to the well with 10 μL of the 5 times diluted sample extract. After 30 min storage at room temperature, without access to light, the absorbance was measured using a plate reader (Multiskan Sky, Thermo Electron Co., St. Louis, MO, USA) at a wavelength of 515 nm. The assays were performed in duplicate for each extract. The antioxidant activity was determined on the basis of a decrease in the absorbance of the radical solution in the presence of the antioxidant and expressed as the Trolox equivalent antioxidant capacity (TEAC) coefficient, corresponding to the concentration of Trolox with the same antioxidant capacity as the tested sample (mg Trolox/g d.m.).

To determine the reduction power (RP) of iron ions by the analyte, 25 μL of the extract, 50 μL of a 1% aqueous solution of potassium ferricyanide, and 75 μL of distilled water were pipetted into the well. The whole was mixed, and then placed in an incubator (INCU-Line ILS 10; VWR, Radnor, PA, USA) at 50 °C. After 20 min, 50 μL of 10% trichloroacetic acid was added. Then, 100 μL of the reaction mixture were taken into an empty well, 100 μL of distilled water and 20 μL of 0.1% iron (III) chloride solution were added, and the whole was mixed. After 10 min, the absorbance at 700 nm was measured against a blank [58]. The RP value is expressed as mg of Trolox.

#### 4.3.7. Determination of Vitamin C Content

The UPLC-PDA method (WATERS Acquity H-Class, Milford, MA, USA) was used to determine the content of L-ascorbic acid [59]. First, 10 mL of cooled extraction reagent (3% metaphosphoric acid and 8% acetic acid) was added to 0.3 g of ground dried material, the solution was stirred with vortex for 5 min, and then centrifuged (2 min, 4350 rpm, 5 °C). The assay was carried out with limited access to light. The solution was filtered using 0.2 μm GHP syringe filters (Acrodisc, Pall Corporation, Port Washington, NY, USA). Next, 1 mL of the solution was added to 1 mL of the eluent, and then injected into the column. A WATERS Acquity UPLC HSS T3 column (2.1 × 100 mm, 1.8 µm, Waters, Ireland) with a BEH C18 pre-column (2.1 × 5 mm, 1.7 µm, Waters, Ireland) was used for separation. The mobile phase flow (Milli-Q water with 0.1% formic acid) was 0.25 mL/min. The column thermostat temperature was 25 °C and the autosampler tempurature was 4 °C. The spectral analysis was performed at a wavelength of 245 nm. The vitamin C content was calculated against a calibration curve prepared for an analytical standard of L-ascorbic acid (0.005–0.100 mg/mL). The analysis was performed in duplicate.

#### 4.3.8. Total Sugar Content (Sucrose, Glucose, Fructose, and Sorbitol)

The sugar content was determined by liquid chromatography [60]. The system consisted of a quadruple pump (Waters 515, Milford, MA, USA), an autosampler (Waters 717, Milford, MA, USA), a column thermostat, and an RI detector (Waters 2414, Milford, MA, USA). First, 0.3 g of the material was poured with MilliQ water at a temperature of 80 °C. The samples were placed in a circular-vibrating shaker and sugars were extracted for 4 h. The solution was centrifuged (5 min, 4350 rpm), filtered through a 0.22 µm hydrophobic PTFE syringe filter (Millex-FG, Millipore, Milford, MA, USA), and 1 µL was injected into the column. Separation was carried out using a 300 × 6.5 mm Waters Sugar Pak I column with a Sugar-Pak pre-column. The assay was carried out with a constant composition of the mobile phase (Milli-Q redistilled water), the flow rate of which was 0.6 mL/min, the detector temperature was 50 °C, and the column temperature was 90 °C. The quantitative analysis was performed on the basis of prepared calibration curves for glucose, fructose, sucrose, and sorbitol standards. The assays were carried out in duplicate.

#### 4.3.9. Fourier Transform Infrared Spectroscopy (FTIR)

FTIR spectra of dried apple tissue were performed in infrared using Cary 630 (Agilent Technologies Inc., Santa Clara, CA, USA) with a single reflection diamond attenuated total reflection ATR [58]. The analysis was carried out at a wavelength in the range of 650–4000 cm^−1^ with a resolution of 4 cm^−1^, with 32 scans of the spectrum. The dried sample was pressed against the crystal with a pressure clamp. Each material was scanned five times. The analysis data was recorded using the MicroLab FTIR software.

### 4.4. Statistical Methods

The experiment was organized using response surface methodology (RSM) with experimental planning of two factors: drying temperature and energy consumption during PEF application, at three levels (Table 3).

The levels of energy consumption during PEF pretreatment applications were selected based on the efficiency of electroporation, determined by measuring the specific electrical conductivity (preliminary studies), while the temperature levels were selected based on the literature. The analysis of the impact of the pulsed electric field intensity and the applied temperature on the quality of the dried samples was assessed on the basis of a one-way ANOVA analysis of variance, and homogeneous groups were determined on the basis of Tukey’s test. Furthermore, in order to determine the impact of PEF pretreatment intensity and hot air drying temperature as well as the interaction of these two factors on the obtained results, a two-factor analysis of variance was performed at the significance level α = 0.05. To conduct the statistical analysis, the Statictica program ver. 13 of the TIBCO company software (Palo Alto, CA, USA) was used.

On the basis of the obtained results, statistical profiles of approximated values of the PEF pretreatment intensity levels and hot air drying temperatures as well as the usability for the drying time and chosen properties (water activity, antioxidant capacity, and total sugars content) were made and presented in Figure 12.

Optimal properties were chosen as the general and most important for obtaining good quality dried material in a short time period. For approximation profiles, the drying time, water activity, and total sugar content were set to obtain lower values, while antioxidant capacity was set to gain the highest values. The projection showed that the best parameters to obtain such dried material would be application of a PEF pretreatment intensity of 3.5 kJ/kg and drying at 70 °C.

## 5. Conclusions

Our results showed that pulsed electric field applied as a pretreatment before the hot air drying process shortened the drying time of the material due to the pores formed that facilitated the evaporation of water from the material. The shortest drying time was obtained when the higher temperature was used. In the case of samples with PEF pretreatment, the shortest drying time was characterized by the material dried at 70 °C and treated using pulsed electric field pretreatment with an intensity of 6 kJ/kg.

Dried materials were characterized by water activity not exceeding 0.6, and the highest value did not exceed 0.331, which was obtained by PEF pretreatment of apple tissue with an electric field intensity of 6 kJ/kg and dried at 60 °C. In addition, electroporation caused a change in the content of chemical compounds in the apple tissue. The application of each electric field intensity caused the degradation of polyphenols from apple tissue compared to samples not subjected to PEF pretreatment. The smallest loss of these substances was noticed in the dried material obtained by applying PEF pretreatment with an electric field intensity of 1 kJ/kg, and then dried at 70 °C. The same trend was observed for flavonoid content and antioxidant capacity. Furthermore, the reducing power investigations confirmed the antioxidant capacity results. PEF pretreatment decreases the reducing power of dried apple tissue compared to the untreated samples. Additionally, ascorbic acid was degraded by pulsed electric field at any level of applied intensity. An increase in energy supplied during PEF application led to an increase in the degradation of vitamin °C in dried apple tissue.

The application of pretreatment did not influence changes in the total sugar content and sorbitol content of dried apple tissue. However, increased fructose and glucose contents were noted with an increase in the drying temperature and the PEF pretreatment electric field intensity. PEF pretreatment did not effect the samples’ FTIR spectra, which were similar for all samples.

The optimization process, as well as statistical profiles of approximated values, showed that the best parameter values for obtaining a high-quality product with a short drying time were the application of PEF pretreatment with an intensity of 3.5 kJ/kg and hot air drying at a temperature of 70 °C.

## Figures and Tables

**Figure 1 molecules-28-02970-f001:**
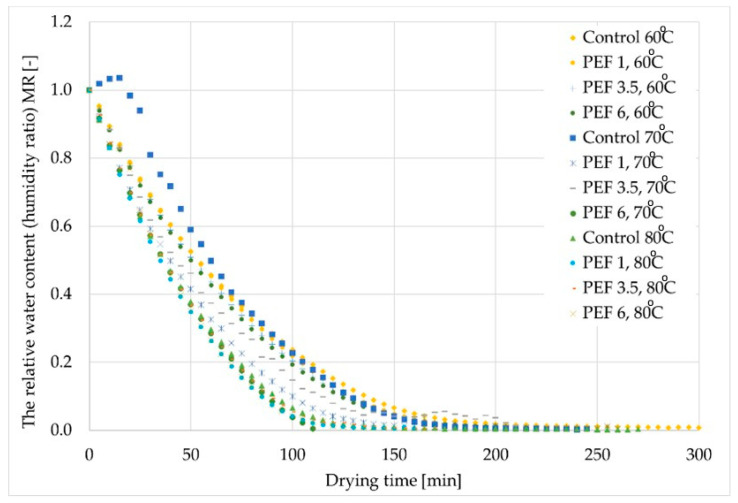
Kinetics of drying apple tissue at temperatures of 60, 70, and 80 °C that has been pretreated with PEF at electric field intensity levels of 1, 3.5, and 6 kJ/kg.

**Figure 2 molecules-28-02970-f002:**
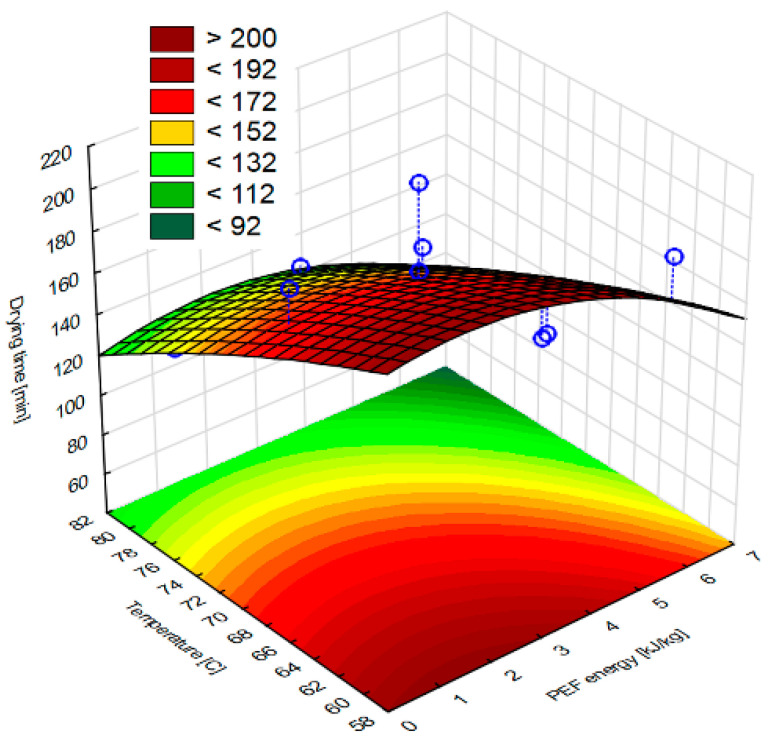
Response surface results for drying time, depending on the pulsed electric field intensity and the temperature obtained in the case of apple tissue subjected to PEF pretreatments with different electric field intensities (1, 3.5, 6 kJ/kg) and hot air drying at different temperatures (60, 70 and 80 °C). Experimental values are marked with points.

**Figure 3 molecules-28-02970-f003:**
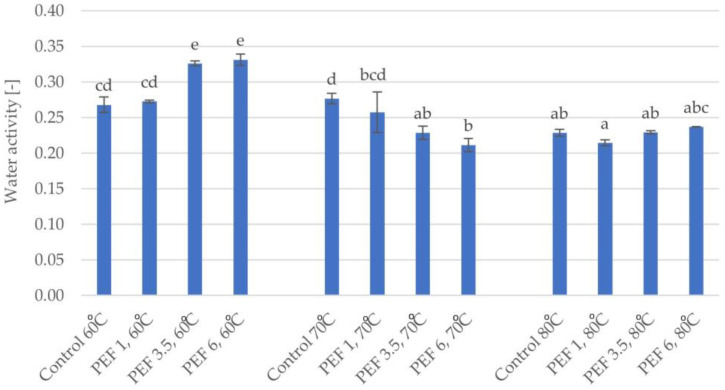
Water activity of dried apple tissue obtained using different values for the parameters of pulsed electric field intensity (1, 3.5, and 6 kJ/kg) and hot air drying temperature (60, 70, and 80 °C). The columns with different letters are significantly different (*p* < 0.05).

**Figure 4 molecules-28-02970-f004:**
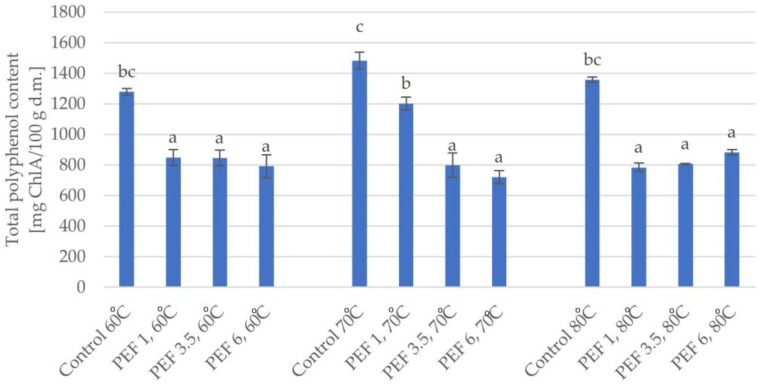
Total polyphenol content of dried apple tissue obtained using different values for the parameters of pulsed electric field intensity (1, 3.5, and 6 kJ/kg) and hot air drying temperature (60, 70, and 80 °C) (*n* = 2). The columns with different letters are significantly different (*p* < 0.05).

**Figure 5 molecules-28-02970-f005:**
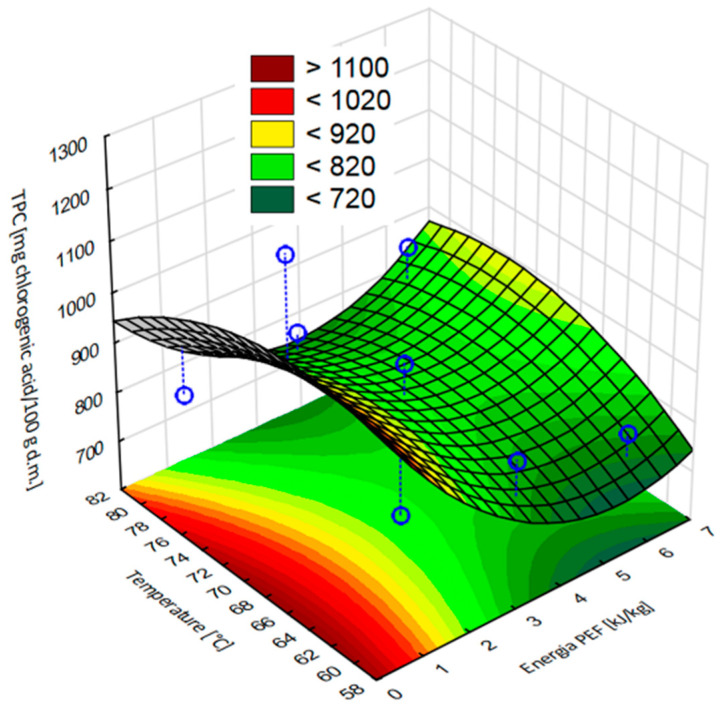
Response surface results of the total polyphenol content, depending on the pulsed electric field intensity and the temperature, obtained in apple tissue subjected to different PEF pretreatment intensities (1, 3.5, and 6 kJ/kg) and hot air drying at different temperatures (60, 70, and 80 °C). Experimental values are marked with points.

**Figure 6 molecules-28-02970-f006:**
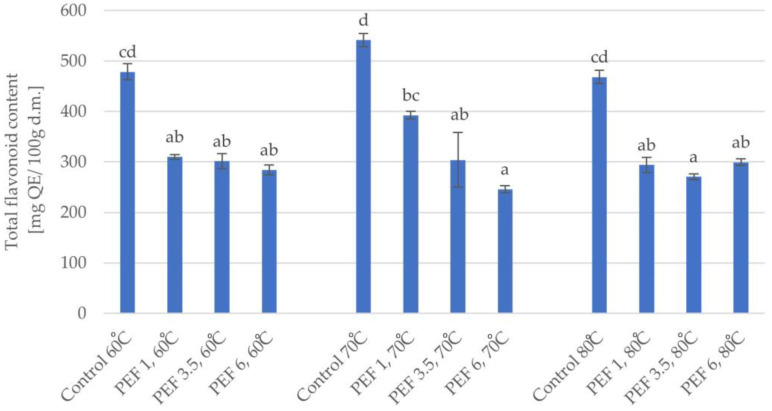
Total flavonoid content of dried apple tissue obtained using different values for the parameters of pulsed electric field intensity (1, 3.5, and 6 kJ/kg) and hot air drying temperature (60, 70, and 80 °C). The columns with different letters are significantly different (*p* < 0.05).

**Figure 7 molecules-28-02970-f007:**
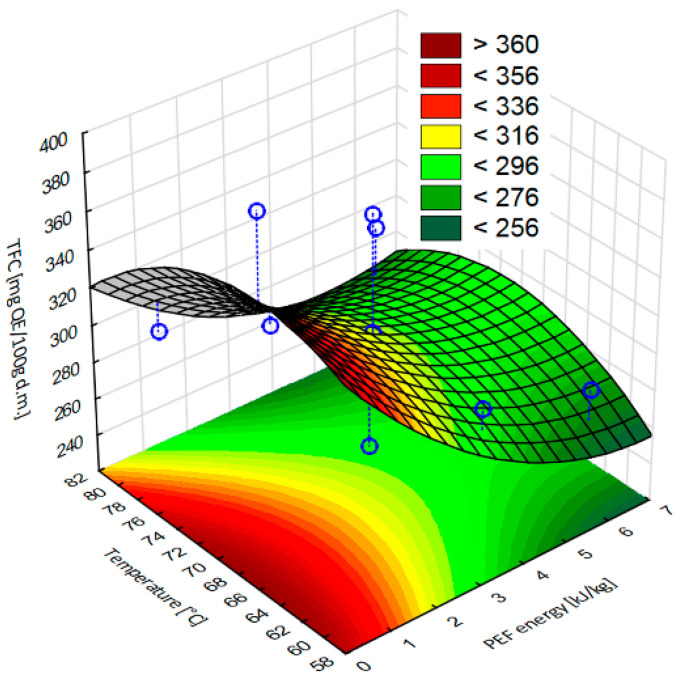
Response surface results of the total flavonoid content, depending on the pulsed electric field intensity and the temperature, obtained in the case of apple tissue subjected to different PEF pretreatment intensities (1, 3.5, and 6 kJ/kg) and hot air drying at different temperatures (60, 70, and 80 °C). Experimental values are marked with points.

**Figure 8 molecules-28-02970-f008:**
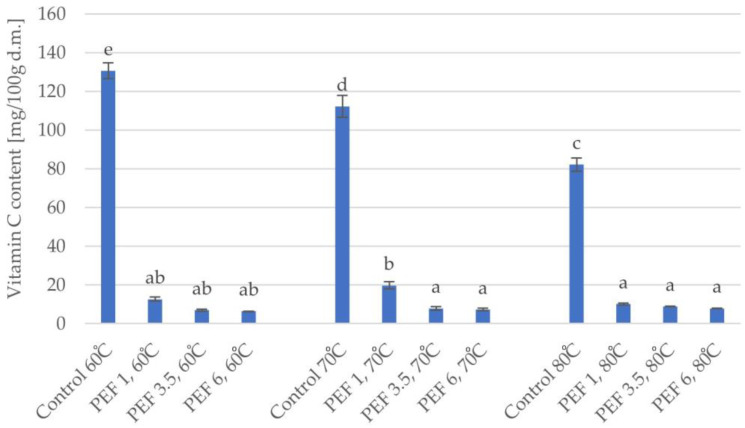
Vitamin C content of dried apple tissue obtained using different values for the parameters of pulsed electric field intensity (1, 3.5, and 6 kJ/kg) and hot air drying temperature (60, 70, and 80 °C). The columns with different letters are significantly different (*p* < 0.05).

**Figure 9 molecules-28-02970-f009:**
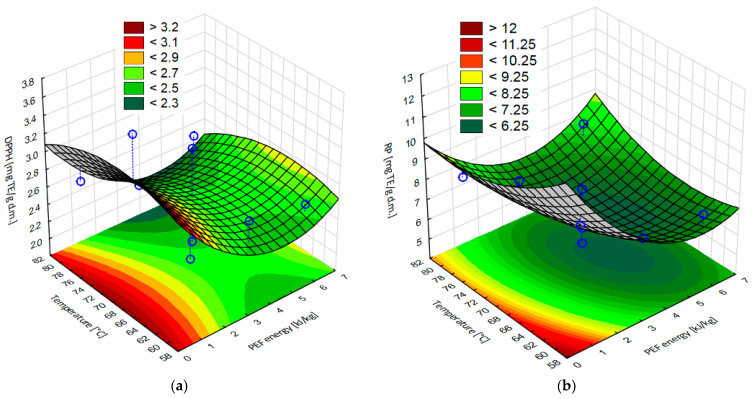
Response surface results of (**a**) the antioxidant capacity with DPPH radical and (**b**) reducing power RP, depending on the pulsed electric field intensity and the hot air drying temperature, obtained in the case of apple tissue subjected to different PEF pretreatment intensities (1, 3.5, and 6 kJ/kg) and hot air drying at different temperatures (60, 70, and 80 °C). Experimental values are marked with points.

**Figure 10 molecules-28-02970-f010:**
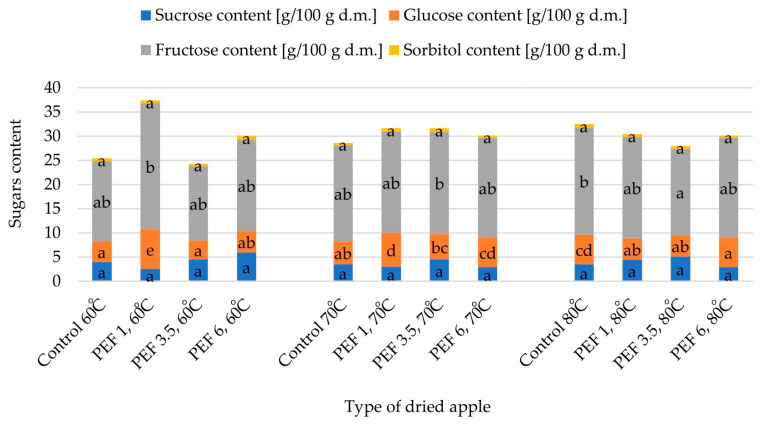
Total sugar content (sucrose, glucose, fructose, and sorbitol) of dried apple tissue obtained using different values for the parameters of pulsed electric field intensity (1, 3.5, and 6 kJ/kg) and hot air drying temperature (60, 70, and 80 °C). The different letters for the one type of sugars shows the significant differences between the samples (*p* < 0.05).

**Figure 11 molecules-28-02970-f011:**
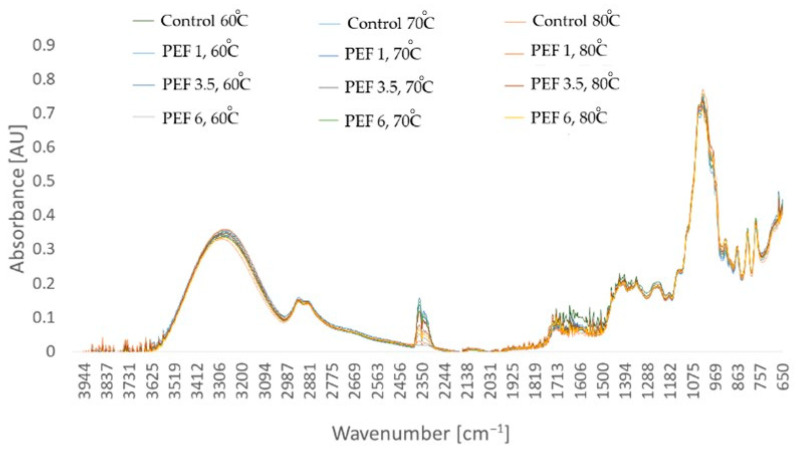
FTIR spectra of dried apple tissue obtained using different values for the parameters of pulsed electric field intensity (1, 3.5, and 6 kJ/kg) and hot air drying temperature (60, 70, and 80 °C). The columns with different letters are significantly different (*p* < 0.05).

**Figure 12 molecules-28-02970-f012:**
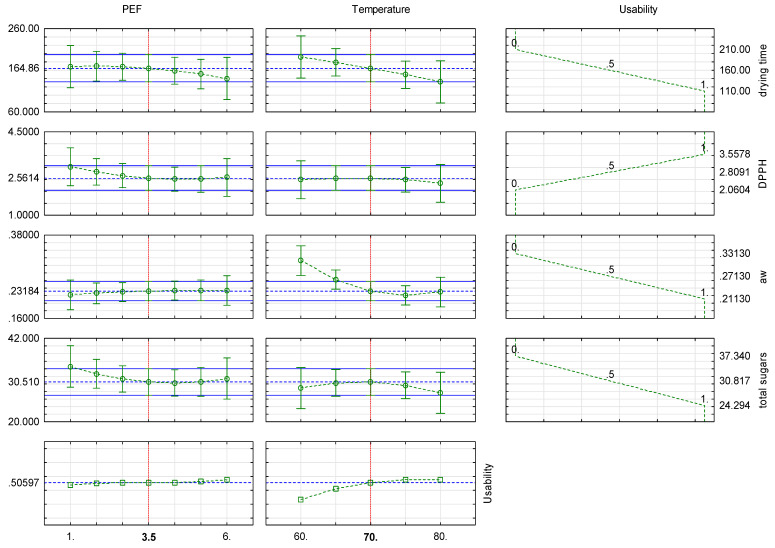
Approximation profiles and usability for dried apple tissue.

**Table 1 molecules-28-02970-t001:** Properties of the fresh apple sample.

Material	Fresh Apple
Water content (%)	16 ± 0.01
Water activity (-)	0.986 ± 0.04
Bioactive compounds	
TPC (mg ChlA/100 g d.m.)	1683 ± 2
TFC (mg QE/100 g d.m.)	715 ± 8
Vitamin C (mg/100 g d.m.)	143.2 ± 5.0
Antioxidant capacity	
AC (mg TE/100 g d.m.)	4.33 ± 0.06
RP (mg TE/100 g d.m.)	20.48 ± 0.72
Sugar content (g/100 g d.m.)	36.71 ± 2.99
Sucrose (g/100 g d.m.)	4.94 ± 0.24
Glucose (g/100 g d.m.)	6.11 ± 0.68
Fructose (g/100 g d.m.)	24.8 ± 2.02
Sorbitol (g/100 g d.m.)	0.86 ± 0.06

**Table 2 molecules-28-02970-t002:** Antioxidant capacity (with DPPH radicals and iron ion reducing power RP) of dried apple tissue obtained using different values for the parameters of pulsed electric field intensity (1, 3.5, and 6 kJ/kg) and hot air drying temperature (60, 70, and 80 °C).

Symbol	Process Parameters		Antioxidant Capacity (AC)	
	Temperature (°C)	PEF (kJ/kg)	DPPH (mg TE/g d.m.)	RP (mg TE/g d.m.)
Control 60 °C	60	-	3.31 ± 0.12 ^abcd*^	14.34 ± 0.99 ^c^
PEF 1, 60 °C	60	1	2.61 ± 0.04 ^abc^	9.42 ± 0.44 ^b^
PEF 3.5, 60 °C	60	3.5	2.73 ± 0.13 ^abcd^	7.40 ± 0.26 ^ab^
PEF 6, 60 °C	60	6	2.61 ± 0.28 ^abc^	7.19 ± 0.17 ^ab^
Control 70 °C	70	-	3.80 ± 0.11 ^d^	17.72 ± 0.14 ^d^
PEF 1, 70 °C	70	1	3.56 ± 0.28 ^cd^	9.64 ± 0.20 ^b^
PEF 3.5, 70 °C	70	3.5	2.51 ± 0.55 ^ab^	6.36 ± 1.40 ^a^
PEF 6, 70 °C	70	6	2.26 ± 0.17^a^	5.35 ± 0.19 ^a^
Control 80 °C	80	-	3.46 ± 0.09 ^bcd^	15.23 ± 0.71 ^cd^
PEF 1, 80 °C	80	1	2.63 ± 0.03 ^abcd^	7.96 ± 0.28 ^ab^
PEF 3.5, 80 °C	80	3.5	2.29 ± 0.15 ^ab^	6.67 ± 0.17 ^ab^
PEF 6, 80 °C	80	6	2.59 ± 0.14 ^abc^	8.06 ± 0.27 ^ab^

SD—standard deviation. *,a,b,c,d—the values in the same column with different letters are significantly different (*p* < 0.05).

**Table 3 molecules-28-02970-t003:** List of dryings, taking into account the temperature and energy supplied during application of the PEF pretreatment—experiment plan.

		Drying Parameters	
Drying Number	Symbol	Temperature (°C)	Energy PEF (kJ/kg)
1	PEF 1, 60 °C	1	60
2	PEF 1, 80 °C	1	80
3	PEF 6, 60 °C	6	60
4	PEF 6, 80 °C	6	80
5	PEF 1, 70 °C	1	70
6	PEF 6, 70 °C	6	70
7	PEF 3.5, 60 °C	3.5	60
8	PEF 3.5, 80 °C	3.5	80
9 (A)	PEF 3.5, 70 °C	3.5	70
10 (B)	PEF 3.5, 70 °C	3.5	70
11 (C)	PEF 3.5, 70 °C	3.5	70
12	Control 60 °C	-	60
13	Control 70 °C	-	70
14	Control 80 °C	-	80

## Data Availability

Not applicable.

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
