# Peer review of "The Influence of Pulsed Electric Field and Air Temperature on the Course of Hot-Air Drying and the Bioactive Compounds of Apple Tissue"

_molecules, 2023, doi:10.3390/molecules28072970_

Round 1

Reviewer 1 Report

The authors explore the effects of pulsed electric field (PEF), as a pretreatment to drying, applied to Gloster apple tissue, on the physical and chemical properties of various compounds. The study found that PEF influences the degradation of bioactive compounds such as polyphenols, flavonoids, and Vitamin C. Total sugar content and sorbitol, however, were not influenced by PEF. The authors also establish optimized values of PEF energy and drying temperature in order to achieve high-quality dried apple in a short drying time.

Figure 1

Please state what is “MR” at the figure caption, and after line 109 where you reference Figure 1.

Why the j_70C sample does not begin from time = 0 min?

Line 126

Typo

Line 181 to Line 193, and Figure 4

You state the results, but can you please give explanation to the observed results? Why is that the PEF application resulted in a slight increase in the content of polyphenols in the apple tissue? Why is that the total polyphenols are less in j_60C than in J_70C? Can you state the number of samples (n = ?) in the caption of Figure 4.

However, in the next paragraph, you state that the use of PEF causes a decrease in the total flavonoid content, and increasing the PEF energy in most cases causes a slight decrease, which is in contrast to the results obtained on the content of polyphenols.

Figure 10

Why is j_60C lower in general to j_PEF_1_60C?

Paragraph at Line 133

It will be interesting to come up with a model (LDA, QDA, etc.) that would predict, with equation, the response variables, the parameters of the PEF and drying that you explore. Moreover, as you state, it will be interesting in this model to see what is the “interaction” of PEF and temperature.

In general, the article is comprehensive and explorational, written in grammatically correct English language. I will recommend the publishing of the manuscript as long as minor revisions are handled.

Author Response

Response to Reviewer 1 Comments

Point 1: The authors explore the effects of pulsed electric field (PEF), as a pretreatment to drying, applied to Gloster apple tissue, on the physical and chemical properties of various compounds. The study found that PEF influences the degradation of bioactive compounds such as polyphenols, flavonoids, and Vitamin C. Total sugar content and sorbitol, however, were not influenced by PEF. The authors also establish optimized values of PEF energy and drying temperature in order to achieve high-quality dried apple in a short drying time.

Response 1: We would like to thank for the comments and your time. The manuscript has been revised based on the comments of the reviewers. All the changes in manuscript were marked on red color.

Point 2: Figure 1 - Please state what is “MR” at the figure caption, and after line 109 where you reference Figure 1.

Response 2: The suggestion was taken into account in the manuscript.

Line 108-110: Figure 1 shows the drying time of apple tissue subjected to different PEF energy and dried at different drying air temperatures in relation to the relative water content (humidity ratio, MR).

Point 3: Why the j_70C sample does not begin from time = 0 min?

Response 3: The suggestion was taken into account in the manuscript.

Point 4: Line 126 Typo

Response 4: The suggestion was taken into account in the manuscript.

Line 124-126: Figure 2. Response surfaces of the drying time, depending on the pulsed electric field energy and temperature obtained in the case of apples subjected to PEF treatment (1, 3.5, 6 kJ/kg) and hot-air drying at different temperatures (60, 70 and 80 °C). Experimental values are marked with points.

Point 5: Line 181 to Line 193, and Figure 4

You state the results, but can you please give explanation to the observed results? Why is that the PEF application resulted in a slight increase in the content of polyphenols in the apple tissue? Why is that the total polyphenols are less in j_60C than in J_70C? Can you state the number of samples (n = ?) in the caption of Figure 4.

However, in the next paragraph, you state that the use of PEF causes a decrease in the total flavonoid content, and increasing the PEF energy in most cases causes a slight decrease, which is in contrast to the results obtained on the content of polyphenols.

Response 5: The suggestion was taken into account in the manuscript. The number of samples (n = 2) were added in the caption of Figure 4.

The slight increase in polyphenol contents with the increase of PEF energy was shown only for samples dried at 80 °C, but differences were statistically insignificant. For samples dried at 80 °C, similar to the samples dried at 60 and 70 °C the PEF pre-treatment caused decrease of polyphenol content, what is in agree with the total flavonoid changes. For most cases PEF energy increase caused statistically insignifican changes in polyphenols content and flavonoid content. Lammerskitten et al. [2019] explained that during PEF-processing, biochemical reactions were occurring, which lead to the formation of new compounds and increase of TPC was for freeze-dried apples, but diffrences were statistically significant.

Point 6: Why is j_60C lower in general to j_PEF_1_60C?

Response 6: Differences are in most cases statistically insignificant (only for glucose content were significant). In manuscript this phenomenon was expained.

Line 519-523: „……..Additionally, the authors suggested that the increase in the glucose content of PEF-treated samples might be caused by the decomposition of carbohydrates in such a way that it can affect other carbohydrates, and also improve the extractability of sugars [43]. Furthermore, PEF treatment of chokeberry juice sources from six different farms caused an increase in the total sugar content in all cases [44].”

Point 7: Paragraph at Line 133

It will be interesting to come up with a model (LDA, QDA, etc.) that would predict, with equation, the response variables, the parameters of the PEF and drying that you explore. Moreover, as you state, it will be interesting in this model to see what is the “interaction” of PEF and temperature.

Response 7: This is good idea for the next manuscript with the physical properties for those samples.

Reviewer 2 Report

Journal: Molecules (ISSN 1420-3049)

Manuscript ID: molecules-2272737

Research Article

The influence of pulsed electric field and air temperature on the course of hot-air drying and bioactive compounds of apple tissue

General comments:

The article in question falls short of the standards required for scientific writing due to a lack of appropriate style and grammar mistakes. While the current version is unsuitable for publication, with proper revisions, it could meet the necessary criteria. To improve the article, the author should focus on developing a clear and concise writing style, avoiding common grammatical errors, and ensuring that all claims are supported by reliable sources. By taking these steps, the article can be transformed into a compelling piece of scientific writing that is suitable for publication.

Specific Comments:

Line number 18: In study there are 3 energies mentioned i.e., 1, 3.5 and 5 kJ/kg but in abstract only two are mentioned. Kindly review this.

Line number 19: Specify if this is apple tissue (surface layer or complete tissue).

Line number 21: You can rewrite this phrase: "The degradation of vitamin C was higher with increase in PEF energy."

Line number 24-26: Rephrase for better understanding.

Line number 32: Try to use "i.e.," instead of dash "-"

Line number 35: “also other advantages

Line number 59: Does not make sense. High content of which compounds or colors? Kindly specify the relation with maintenance of final material as a result of pretreatment.

Line number 60-62: Rephrase this line for better readability.

Line number 69: Fix grammar “interact

Line number 86: Specify intensity limit for PEF which is considered too intense.

Results: Is the prefix "j_" really required if this is constant in every case? I suggest changing treatment labels for better readbility for users as in this case it seems like the treatment labels are generated from computer. They can be renamed to

Control 60C

PEF 1, 60C

PEF 3.5, 60C

PEF 6, 60C

and so on.

Line number 126: Change kJ/k to kJ/kg (Fix this everywhere)

Line number 130: Fix spellings: results

Table 2: Fix spellings: Water content

Line number 141-142: “it can be concluded that in most cases, there were significant differences..

Line number 143: “As the response surfaces of the water…”

Line number 151: “apple tissue were…”

Line number 160: Fix “obserwed” to “observed”.

Line number 167: Fragment sentence, does not make sense

Line number 188: Fix approx. to approximately

Line number 197: “the total polyphenols content was..”

Line number 234: “that applied pre-treatment has..”

Line number 283: Fix “significan” to significant

Line number 284: Fix furtheremore to furthermore

Line number 290: There are two “and”

Line number 320: Fix “sorbital” to “sorbitol”

Line number 442: Fix “Antioxidants” to “Antioxidants”

Line number 557: Materials and Methods: This whole portion should be mentioned before the results and discussion portion.

Line number 562: Fix “aplles” to “apples” and “presentend” to “presented”

Line number 609: Space between “minutes.Such”

Line number 713: “Those properties were..”

Line number 714: Fix “obtaning” to “obtaining”

Line number 716: Fix “higerst” to “highest”

Line number 729: Fix “exiceeded” to “exceeded”

Line number 733: “The smallest loss of these substances was noticed…”

Some more studies can be added to discussion section, following are some suggestions:

·         Lammerskitten, A., Wiktor, A., Siemer, C., Toepfl, S., Mykhailyk, V., Gondek, E., .. & Parniakov, O. (2019). The effects of pulsed electric fields on the quality parameters of freeze-dried apples. Journal of Food Engineering252, 36-43.

·         Arshad, R. N., Abdul-Malek, Z., Munir, A., Buntat, Z., Ahmad, M. H., Jusoh, Y. M., .. & Aadil, R. M. (2020). Electrical systems for pulsed electric field applications in the food industry: An engineering perspective. Trends in food science & technology104, 1-13.

·         Wiktor, A., Iwaniuk, M., Åšledź, M., Nowacka, M., Chudoba, T., & Witrowa-Rajchert, D. (2013). Drying kinetics of apple tissue treated by pulsed electric field. Drying Technology31(1), 112-119.

·         Bazhal, M. I., Lebovka, N. I., & Vorobiev, E. (2001). Pulsed electric field treatment of apple tissue during compression for juice extraction. Journal of Food Engineering50(3), 129-139.

·         Roobab, U., Abida, A., Chacha, J. S., Athar, A., Madni, G. M., Ranjha, M. M. A. N., .. & Trif, M. (2022). Applications of innovative non-thermal pulsed electric field technology in developing safer and healthier fruit juices. Molecules27(13), 4031.

Author Response

Response to Reviewer 2 Comments

Point 1: General comments:

The article in question falls short of the standards required for scientific writing due to a lack of appropriate style and grammar mistakes. While the current version is unsuitable for publication, with proper revisions, it could meet the necessary criteria. To improve the article, the author should focus on developing a clear and concise writing style, avoiding common grammatical errors, and ensuring that all claims are supported by reliable sources. By taking these steps, the article can be transformed into a compelling piece of scientific writing that is suitable for publication.The authors explore the effects of pulsed electric field (PEF), as a pretreatment to drying, applied to Gloster apple tissue, on the physical and chemical properties of various compounds. The study found that PEF influences the degradation of bioactive compounds such as polyphenols, flavonoids, and Vitamin C. Total sugar content and sorbitol, however, were not influenced by PEF. The authors also establish optimized values of PEF energy and drying temperature in order to achieve high-quality dried apple in a short drying time.

Response 1: We would like to thank for the comments and your time. The manuscript has been revised based on the comments of the reviewers. All the changes in manuscript were marked on red color.

Point 2: Specific Comments:

  1. Line number 18: In study there are 3 energies mentioned i.e., 1, 3.5 and 5 kJ/kg but in abstract only two are mentioned. Kindly review this.

Response 2: The suggestion was taken into account in the manuscript.

Line 17-18: „Dried samples were obtained with a combination of pre-treatment (PEF) with the energies of 1, 3.5, and 6 kJ / kg and subsequent drying at 60, 70, and 80°C.”

Point 3: 2. Line number 19: Specify if this is apple tissue (surface layer or complete tissue).

Response 3: Answer: The suggestion was taken into account in the manuscript.

Line 14-17: “Thus, the aim of the study was to investigate the effect of the pulsed electric field as pre-treatment, and air temperature on the course of hot-air drying and selected chemical properties of the Gloster variety apple complete tissue.”

Point 4: 3. Line number 21: You can rewrite this phrase: "The degradation of vitamin C was higher with increase in PEF energy."

Response 4: The suggestion was taken into account in the manuscript.

Line 21-22: „The degradation of vitamin C was higher with the increase in PEF energy.”

Point 5: 4. Line number 24-26: Rephrase for better understanding.

Response 5: The suggestion was taken into account in the manuscript.

Line 23-25: “According to optimization process and statistical profiles of approximated values the optimal parameters to achieve a high-quality dried apple in a short drying time are an application PEF of 3.5 kJ/kg and drying at a temperature of 70°C.”

Point 6: 5. Line number 32: Try to use "i.e.," instead of dash "-"

Response 6: The suggestion was taken into account in the manuscript.

Point 7: 6. Line number 35: “also other advantages”

Response 7: The suggestion was taken into account in the manuscript.

Line 33-36: “Despite also other advantages of drying food, there are also disadvantages, such as changes in sensory characteristics, color, and degradation of nutrients susceptible to high temperatures, which is associated with the deterioration of the quality of the final product”

Point 8: 7. Line number 59: Does not make sense. High content of which compounds or colors? Kindly specify the relation with maintenance of final material as a result of pretreatment.

Response 8: The suggestion was taken into account in the manuscript.

Line 56-59: “In addition, the use of preliminary treatments, with properly selected parameters, allows for to maintenance of the good quality of the final material, e.g., with a high content of certain nutritive compounds (anthocyanin, total phenolics, vitamin C, and antioxidant activity).”

Point 9: 8. Line number 60-62: Rephrase this line for better readability.

Response 9: The suggestion was taken into account in the manuscript.

Line 59-63: “The pulsed electric field (PEF) as a pre-treatment is a type of non-thermal technology that is becoming more and more popular. Under the influence of the applied electric field of appropriate intensity, pores have formed that increase the permeability of the membrane and facilitate the removal of water from the material during drying [8].”

Point 10: 9. Line number 69: Fix grammar “interact”

Response 10: The suggestion was taken into account in the manuscript.

Line 68-69: “………..cell membrane, which interact with each other,…..”

Point 11: 10. Line number 86: Specify intensity limit for PEF which is considered too intense

Response 11: The suggestion was taken into account in the manuscript.

Line 84-87: „Pre-treatment in the form of a pulsed electric field with appropriate parameters has a positive effect on the bioactive components of the dried material, while too intense PEF application (3 kJ/kg used in presented investigations) may have a negative impact on the final quality of the dried material [15].”

Point 12: 11. Results: Is the prefix "j_" really required if this is constant in every case? I suggest changing treatment labels for better readbility for users as in this case it seems like the treatment labels are generated from computer. They can be renamed to 

Control 60C

PEF 1, 60C

PEF 3.5, 60C

PEF 6, 60C

and so on.

Response 12: The suggestion was taken into account in the manuscript.

Point 13: 12. Line number 126: Change kJ/k to kJ/kg (Fix this everywhere)

Response 13: The suggestion was taken into account in the manuscript.

Point 14: 13. Line number 130: Fix spellings: results

Response 14: The suggestion was taken into account in the manuscript.

Point 15: 14. Table 2: Fix spellings: Water content

Response 15: The suggestion was taken into account in the manuscript.

Point 16: 15. Line number 141-142: “it can be concluded that in most cases, there were significant differences..”

Response 16: The suggestion was taken into account in the manuscript.

Point 17: 16. Line number 143: “As the response surfaces of the water…”

Response 17: The suggestion was taken into account in the manuscript.

Point 18: 17.  Line number 151: “apple tissue were…”

Response 18: The suggestion was taken into account in the manuscript.

Point 19: 18.  Line number 160: Fix “obserwed” to “observed”.

Response 19: The suggestion was taken into account in the manuscript.

Point 20: 19.  Line number 167: Fragment sentence, does not make sense

Response 20: The suggestion was taken into account in the manuscript.

Line 166-168: “When drying at 70 °C, it can be concluded that, as in the case of drying at 60 °C, the use of pre-treatment reduced the number of polyphenols compared to the control sample (dried apples obtained without pre-treatment).”

Point 21: 20.  Line number 188: Fix approx. to approximately

Response 21: The suggestion was taken into account in the manuscript.

Point 22: 21.  Line number 197: “the total polyphenols content was..”

Response 22: The suggestion was taken into account in the manuscript.

Point 23: 22.  Line number 234: “that applied pre-treatment has..”

Response 23: The suggestion was taken into account in the manuscript.

Point 24: 23.  Line number 283: Fix “significan” to significant

Response 24: The suggestion was taken into account in the manuscript.

Point 25: 24.  Line number 284: Fix furtheremore to furthermore

Response 25: The suggestion was taken into account in the manuscript.

Point 26: 25.  Line number 290: There are two “and”

Response 26: The suggestion was taken into account in the manuscript.

Point 27: 26.  Line number 320: Fix “sorbital” to “sorbitol”

Response 27: The suggestion was taken into account in the manuscript.

Point 28: 27.  Line number 442: Fix “Antioxidants” to “Antioxidants”

Response 28: The suggestion was taken into account in the manuscript.

Point 29: 28.  Line number 557: Materials and Methods: This whole portion should be mentioned before the results and discussion portion.

Response 29: According to the “Instructions for authors” this portion is after “Discussion” “Research Manuscript Sections

  • Introduction:…
  • Results:...
  • Discussion:….
  • Materials and Methods:…
  • Conclusions:…”

Point 30: 29.  Line number 562: Fix “aplles” to “apples” and “presentend” to “presented”

Response 30: The suggestion was taken into account in the manuscript.

Point 31: 30.  Line number 609: Space between “minutes.Such”

Response 31: The suggestion was taken into account in the manuscript.

Point 32: 31.  Line number 713: “Those properties were..”

Response 32: The suggestion was taken into account in the manuscript.

Point 33: 32.  Line number 714: Fix “obtaning” to “obtaining”

Response 33: The suggestion was taken into account in the manuscript.

Point 34: 33.  Line number 716: Fix “higerst” to “highest”

Response 34: The suggestion was taken into account in the manuscript.

Point 35: 34.  Line number 729: Fix “exiceeded” to “exceeded”

Response 35: The suggestion was taken into account in the manuscript.

Point 36: 35.  Line number 733: “The smallest loss of these substances was noticed…”

Response 36: The suggestion was taken into account in the manuscript.

Point 37: 36.  Some more studies can be added to discussion section, following are some suggestions:

Lammerskitten, A., Wiktor, A., Siemer, C., Toepfl, S., Mykhailyk, V., Gondek, E., .. & Parniakov, O. (2019). The effects of pulsed electric fields on the quality parameters of freeze-dried apples. Journal of Food Engineering, 252, 36-43.

Arshad, R. N., Abdul-Malek, Z., Munir, A., Buntat, Z., Ahmad, M. H., Jusoh, Y. M., .. & Aadil, R. M. (2020). Electrical systems for pulsed electric field applications in the food industry: An engineering perspective. Trends in food science & technology, 104, 1-13.

Wiktor, A., Iwaniuk, M., Śledź, M., Nowacka, M., Chudoba, T., & Witrowa-Rajchert, D. (2013). Drying kinetics of apple tissue treated by pulsed electric field. Drying Technology, 31(1), 112-119.

Bazhal, M. I., Lebovka, N. I., & Vorobiev, E. (2001). Pulsed electric field treatment of apple tissue during compression for juice extraction. Journal of Food Engineering, 50(3), 129-139.

Roobab, U., Abida, A., Chacha, J. S., Athar, A., Madni, G. M., Ranjha, M. M. A. N., .. & Trif, M. (2022). Applications of innovative non-thermal pulsed electric field technology in developing safer and healthier fruit juices. Molecules, 27(13), 4031.

Response 37: The suggestion was taken into account in the manuscript.

Round 2

Reviewer 2 Report

Ok.